# One-Point Statistics Matter in Extended Cosmologies

Alex Gough *[ID] and Cora Uhlemann [ID]

School of Mathematics, Statistics and Physics, Newcastle University, Herschel Building,
Newcastle upon Tyne NE1 7RU, UK; cora.uhlemann@newcastle.ac.uk
*   Correspondence: a.gough2@newcastle.ac.uk

**Abstract:** The late universe contains a wealth of information about fundamental physics and gravity, wrapped up in non-Gaussian fields. To make use of as much information as possible, it is necessary to go beyond two-point statistics. Rather than going to higher-order $N$-point correlation functions, we demonstrate that the probability distribution function (PDF) of spheres in the matter field (a one-point function) already contains a significant amount of this non-Gaussian information. The matter PDF dissects different density environments which are lumped together in two-point statistics, making it particularly useful for probing modifications of gravity or expansion history. Our approach in Cataneo et al. 2021 extends the success of Large Deviation Theory for predicting the matter PDF in ΛCDM in these "extended" cosmologies. A Fisher forecast demonstrates the information content in the matter PDF via constraints for a *Euclid*-like survey volume combining the 3D matter PDF with the 3D matter power spectrum. Adding the matter PDF halves the uncertainties on parameters in an evolving dark energy model, relative to the power spectrum alone. Additionally, the matter PDF contains enough non-linear information to substantially increase the detection significance of departures from General Relativity, with improvements up to six times the power spectrum alone. This analysis demonstrates that the matter PDF is a promising non-Gaussian statistic for extracting cosmological information, particularly for beyond ΛCDM models.

**Keywords:** cosmology; theory; large-scale structure of the Universe; analytical methods

## 1. Introduction

In recent decades, cosmology has moved solidly into a data-driven science. The current standard model of cosmology, called ΛCDM, consists of a cosmological constant as the dark energy component (Λ), and cold (non-relativistic) dark matter (CDM) as its principle components.

The parameters of the ΛCDM model are most tightly constrained currently by experiments measuring the temperature anisotropies and polarisation in the cosmic microwave background (CMB) (for example, the *Planck* measurements in [1]). However, while CMB data is very valuable in extracting cosmological information, in the push to sub-percent measurements of standard cosmological parameters, and in testing non-standard cosmologies, the large-scale structure (LSS) of the universe is the most promising complementary tool.

The principle advantage of LSS data is that it is three dimensional, tracing a history of how cosmic structure evolves over time, since the snapshot of the CMB. By counting the Fourier modes available, one can expect 1–2 orders of magnitude improvement on constraints from LSS data (see, e.g., [2]). In particular, the large-scale structure provides a window to the expansion history of the universe, which makes it an exciting probe of dynamical dark energy and modifications to gravity. These extensions to the standard cosmology are one of the principle science goals of current and upcoming missions like *Euclid* [3], LSST [4], and DESI [5]. These extensions to the standard ΛCDM model would represent new fundamental physics, and could resolve certain current observational tensions which ΛCDM is strained to explain (recently reviewed in, e.g., [6–8]).

However, extracting information from the late universe via large-scale structure is non-trivial for several reasons. The first and most relevant for this work is that the late universe is statistically much more complex than the universe at the time of the CMB. Extracting cosmological information is always done on a statistical basis, treating observables such as a field of galaxy positions or shear lensing maps as single realisations of a random field. Gaussian random fields are completely characterised by their two-point correlation function (or its Fourier counterpart the power spectrum), as all higher-order correlations functions can be written as sums of the two-point function via Wick's/Isserlis' theorem. The CMB has been measured to be a near-perfect Gaussian random field [9], and so measurement of the power spectrum is sufficient to quantify all information content in the CMB. However, as gravitational collapse is a non-linear process, the statistics of the late time density are not also Gaussian, as the non-linearity in the mapping from initial to final densities sources non-trivial higher statistics as information "leaks out" of the power spectrum. It is therefore crucial to determine statistics beyond the power spectrum which can recapture this non-linear information in an efficient and theoretically tractable way. Other reasons why extracting information from LSS data is difficult come down to issues of modelling dynamics on non-linear scales (often side-stepped by running *N*-body simulations, which are expensive and have their own host of non-trivialities), and on a variety of systematic effects in observations.

## 2. Methods

### 2.1. The Matter PDF in Spheres from Large Deviation Theory

This work focuses on a simple choice of non-Gaussian statistic, namely the probability distribution function (PDF) of matter density in spheres. The matter PDF can be straightforwardly calculated from the density field of a cosmological *N*-body simulation by looking at the distribution of the matter field smoothed with a spherical top-hat filter on the scale of interest.

The work of several papers [10–14] provide an analytic framework for predicting the matter PDF in spheres in the mildly non-linear regime. This method relies on large deviations theory (LDT) where the driving parameter is the non-linear variance, $\sigma_{\rm NL}^2$, of the matter field. This formalism therefore remains valid at redshifts $z$ and for spheres of radius $R$ where $\sigma_{\rm NL}^2(z, R) < 1$.

For Gaussian initial conditions, the PDF, $\mathcal{P}^{\rm lin}(\delta_{\rm L})$, of the linear matter density contrast, $\delta_{\rm L}$, in a sphere of radius $r$ is a Gaussian distribution with width given by the linear variance at scale $r$ and redshift $z$

$$\mathcal{P}_{r,z}^{\rm lin}(\delta_L) = \sqrt{\frac{\Psi_{r,z}^{\rm lin\prime\prime}(\delta_{\rm L})}{2\pi}} \exp\left[-\Psi_{r,z}^{\rm lin}(\delta_{\rm L})\right], \quad \Psi_{r,z}^{\rm lin}(\delta_L) = \frac{\delta_{\rm L}^2}{2\sigma_{\rm L}^2(r, z)}. \tag{1}$$

The linear variance on scale $r$ is given by an integral over the linear power spectrum $P_{\rm L}$ with a spherical top-hat filter in position space

$$\sigma_{\rm L}^2(r, z) = \int \frac{{\rm d}k}{2\pi^2} P_{\rm L}(k, z) k^2 W_{\rm 3D}^2(kr), \quad W_{\rm 3D}^2(k) = 3\sqrt{\frac{\pi}{2}} \frac{J_{3/2}(k)}{k^{3/2}}, \tag{2}$$

where $W_{\rm 3D}(k)$ is the Fourier transform of the 3D spherical top-hat filter, and $J_{3/2}(k)$ is the Bessel function of the first kind of order $3/2$.

The function $\Psi$ in Equation (1) is related to the *rate function* in the context of LDT. The key result from the LDT formalism allows us to relate the rate function of the linear density to the non-linear density, which provides the exponential dependence of the final PDF. Generally this LDT result is called the contraction principle and relates the rate function of different random variables. In the cosmological case, since large deviations are exponentially unlikely and the matter PDF is computed for spherically symmetric cells, the most likely mapping from linear to non-linear densities should be dominated by spherical collapse, $\delta_L \to \rho_{\rm SC}(\delta_L)$. Combining this with mass conservation in spheres (which

relates the initial scale, $r$, to the final scale, $R$, by $r = R\rho^{1/3}$) leads to the final decay-rate function $\Psi_{R,z}$ of the non-linear density

$$\Psi_{R,z}(\rho) = \frac{\sigma_L^2(R,z)}{\sigma_L^2(R\rho^{1/3},z)} \frac{\delta_L^{SC}(\rho)^2}{2\sigma_{NL}^2(R,z)}. \tag{3}$$

The LDT model then predicts the matter PDF in spheres is given by

$$\mathcal{P}_{R,z}(\rho) \propto \exp(-\Psi_{R,z}(\rho)), \tag{4}$$

where the precise prefactor can be determined by a more detailed analysis (see Equation (5a,b) from [15]).

For a standard $\Lambda$CDM universe, there are only three quantities needed for this theoretical model of the matter PDF (through the decay-rate function in Equation (3)):

(i)　The time- and scale-dependence of the linear variance $\sigma_L^2(r,z)$.

(ii)　The non-linear variance of the log-density $\sigma_{\ln\rho,NL}^2(R,z)$.

(iii)　The mapping between linear and final densities in spheres, which is taken to be spherical collapse $\delta_L \mapsto \rho_{SC}(\delta_L)$ (or its inverse $\delta_L^{SC}(\rho)$).

Figure 1 shows the success of this LDT model against measured PDFs from the Quijote simulations [16], as well as in comparison to a common log-normal phenomenological model (dashed lines).

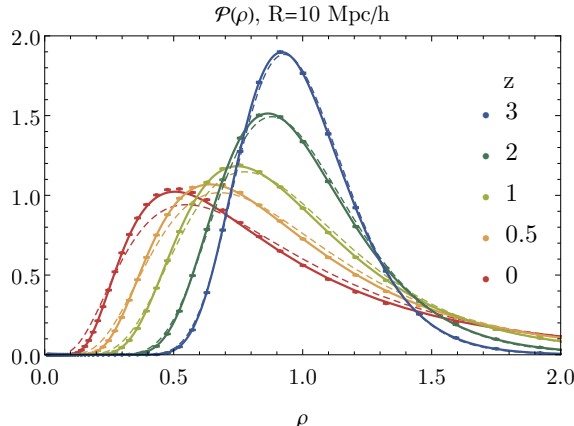

**Figure 1.** The LDT model for the matter PDF (solid lines) compared to measured PDFs from the Quijote simulations (points). The LDT model remains more accurate than a log-normal approximation with the measured variance (dashed lines) on small scales and at late times.

*2.2. Extended Cosmologies*

Due to the significant non-Gaussian information in the matter PDF and the success of this LDT formalism in $\Lambda$CDM, we modify this framework to analyse cosmologies with non-GR theories of gravity or dynamical theories of dark energy beyond a cosmological constant. Collectively we will refer to either modified gravity (MG) or dark energy (DE) models as *extended* cosmologies.

The dark energy model considered was a simple parametrisation of an evolving dark energy, called the $w_0w_a$CDM model. This cosmology is still described by a smooth dark energy and General Relativity (GR), but with dark energy equation of state given by [17,18]

$$w(a) = w_0 + w_a(1 - a), \tag{5}$$

where $\{w_0, w_a\}$ are new phenomenological parameters with $w_0 = -1$, $w_a = 0$ corresponding to the cosmological constant.

For the theories of modified gravity, we considered Hu-Sawiki $f(R)$ gravity [19] and the normal branch of DGP braneworld gravity which acts as an additional smooth dark energy component [20]. The strength of the deviation from GR gravity in these theories is quantified by the parameters $f_{R0}$ and $\Omega_{rc}$ respectively. In moving to an extended cosmology, all three of the ingredients outlined in the previous section in principle need updating: the linear variance, the non-linear variance of the log-density, and the mapping between initial and final densities.

Updating the linear variance is straightforward, simply requiring integrating the modified linear power spectrum which can be achieved by the ratio of linear growth factors. This is achieved using a novel technique for emulating the response of the $\Lambda$CDM power spectrum to MG/DE effects as described in [21]. The non-linear variance can either be measured from a set of simulations in the extended cosmology, or can be well approximated by a phenomenological log-normal rescaling (equation (14) from [22]) applied to a reference cosmology.

The mapping from initial to final densities is trickiest for extended cosmologies, especially in the case of scale dependent modified gravity. However, in [15] we show that when restricted to mildly non-linear scales ($R \gtrsim 10$ Mpc/h) we can use the spherical collapse mapping for an Einstein de Sitter cosmology, modified just by the difference in linear growth factors between the $\Lambda$CDM and extended cosmology.

### 2.3. Simulations and Model Validation

In [15], we validated the predicted matter PDFs against a suite of modified gravity and dark energy simulations, and found them to be accurate to 2% over the range of densities used in the final analysis. All PDFs and the analysis presented here are based on the LDT model, calculated using pyLDT (https://github.com/mcataneo/pyLDT-cosmo, accessed on 4 December 2021) a modularised and user-friendly Python code that takes advantage of the PyJulia interface for computationally intensive tasks. The PDFs used in this paper were generated using version 0.4.9 of pyLDT.

## 3. Results

The following results summarise the analysis presented in Cataneo et al. 2021 [15].

### 3.1. Matter PDFs in Extended Cosmologies

Figure 2 shows the matter PDF in $10 \, h^{-1}$ Mpc spheres for the three different theories of gravity considered, $\Lambda$CDM, $f(R)$, and DGP. Generically, introducing modified gravity will change both the width and the shape of the PDF (c.f. Figure 3 of [15]). Since $\sigma_8$ sets the width of the PDF, normalising the cosmologies to have the same $\sigma_8$ at redshift 0 allows us to isolate the distinct features of modified gravity on the PDF, as done in Figure 2.

By normalising the width of the PDF at redshift 0, we see a residual difference in the shape of the matter PDF, as well as a distinct redshift dependence. These differences in shape and redshift dependence (as well as a difference in scale dependence not shown in Figure 2) are what allows the PDF to break degeneracies between standard cosmological parameters and modified gravity. The effect of an evolving dark energy on the matter PDF is similar to that of DGP gravity, entering mostly by modifying the expansion history.

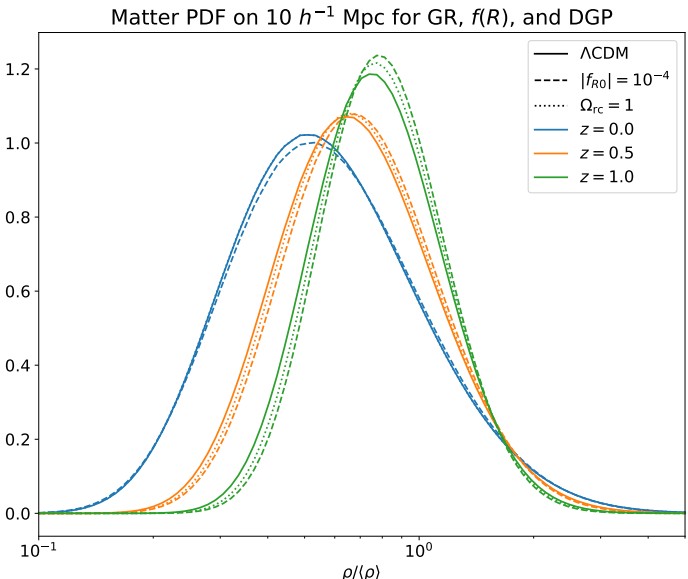

**Figure 2.** Comparison of the matter PDF in 10 h$^{-1}$ Mpc spheres in different theories of gravity. The cosmological parameters are chosen such that the clustering amplitude $\sigma_8$ is the same in all cases at redshift 0. This normalises the overall width of the PDF. The resulting difference in tilt and redshift dependence is due to the change to gravity.

### 3.2. Forecasting Constraining Power with the Fisher Formalism

To forecast errors on a set of cosmological parameters, $\vec{\theta}$, we make use of the Fisher matrix formalism. This formalism provides constraints on $\vec{\theta}$ under the assumption that the likelihood is approximated by a multivariate Gaussian distribution.

Given a (set of) summary statistics arranged in a data vector, $\vec{S}$ with components $S_\alpha$, and the covariance matrix between those summary statistics, $\mathbf{C}_d$, the components of the Fisher matrix, $F$, are defined as (assuming that the data covariance matrix is independent of cosmological parameters)

$$F_{ij} = \sum_{\alpha,\beta} \frac{\partial S_\alpha}{\partial \theta_i} (\mathbf{C}_d^{-1})_{\alpha\beta} \frac{\partial S_\beta}{\partial \theta_j}. \tag{6}$$

Assuming the likelihood of our observed statistics given our cosmological parameters is well approximated by a multivariate Gaussian, the parameter covariance matrix $\mathbf{C}_p(\vec{\theta})$ (encoding the expected parameter constraints along with parameter degeneracies) is given by the inverse of the Fisher matrix. To obtain constraints by marginalising over a subset of parameters, one can simply select the appropriate elements of the parameter covariance matrix. In particular, this implies that the fully marginalised constraint on a single parameter, $\theta_i$, is given by

$$\sigma[\theta_i] = \sqrt{(F^{-1})_{ii}}. \tag{7}$$

For this analysis, we used three different data vectors. These are the PDF alone, the matter power spectrum alone, and a stacked data vector which combines both the PDF and the matter power spectrum. We combined information from three redshifts, $z = 0, 0.5, 1$. The data covariance was measured from the fiducial $\Lambda$CDM runs of the Quijote suite of simulations [16] subsequently rescaled to correspond to a *Euclid*-like survey volume of 20 (Gpc/h)$^3$ , for details see Cataneo et al. 2021 [15]. This covariance matrix encodes statistical errors in the matter PDF and power spectrum, as well as their correlation in the combined data vector. For the matter PDF, we combined three sizes of spheres (10, 15, 20 Mpc/h), while for the matter power spectrum, we included information up to

$k_{max} = 0.2$ h/Mpc. This maximum wavenumber was chosen to ensure that theoretical predictions of the power spectrum are consistent with those measured from simulations, see Appendix C of [15].

### 3.3. Response of the PDF to Changes in Cosmological Parameters

Figures 3 and 4 illustrate how the matter PDF depends on standard parameters related to structure formation $\Omega_m$ and $\sigma_8$, as well as the parameters extending $\Lambda$CDM, namely $|f_{R0}|$, $w_0$, and $w_a$ ($\Omega_{rc}$ for DGP gravity is omitted here to avoid cluttering the plots). This more directly quantifies how well degeneracies in these parameters can be broken, and allows us to quantify the heuristic understanding gained by Figure 2. These derivatives directly enter into the Fisher constraints via Equation (6).

Notice that the effect of $\Omega_m$ on the matter PDF can easily be disentangled from the other parameters in both the DE and MG case, as the matter PDF is sensitive to $\Omega_m$ only through its skewness and the linear growth factor, $D(z)$ [22].

In $f(R)$ gravity, we can expect the most degeneracy breaking, and therefore better constraints on $f_{R0}$ than $\Omega_{rc}$ or $\{w_0, w_a\}$ for two main reasons which can be seen in Figure 3. The first is that the $f_{R0}$ derivative has a different shape from the $\sigma_8$ derivative, showing up as an additional skewness owing to the scale dependent fifth force. This, combined with the fact that the $f_{R0}$ derivatives are non-zero at $z = 0$ (unlike in DGP and $w_0 w_a$CDM), allows more information to be extracted from the non-linear regime. While DGP gravity is not shown in Figure 3, its effect is very similar to that of dark energy shown in Figure 4 (as can be seen in Figure 11 from [15]). Figure 4 shows that at fixed scale and redshift, the response of the PDF to $w_0$ or $w_a$ is very similar in shape to the response to $\sigma_8$. For this reason, in the cases of dark energy and DGP gravity, the degeneracy is mainly broken by a difference in the redshift (and scale) dependence.

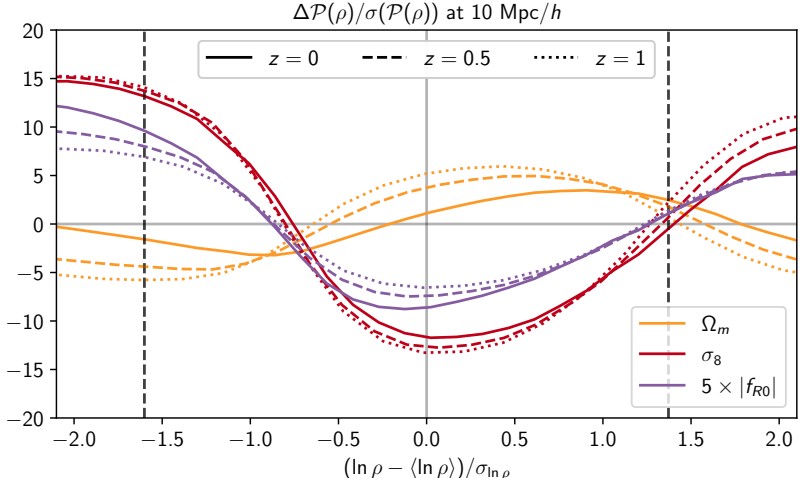

**Figure 3.** Derivatives of the matter PDF in an $f(R)$ modified gravity scenario. The $f(R)$ gravity parameter, $|f_{R0}|$, can be distinguished from $\sigma_8$ by both its redshift dependence and its additional skewness. $\Omega_m$ can be disentangled from both by its different effect on skewness, and its effect on the linear growth factor.

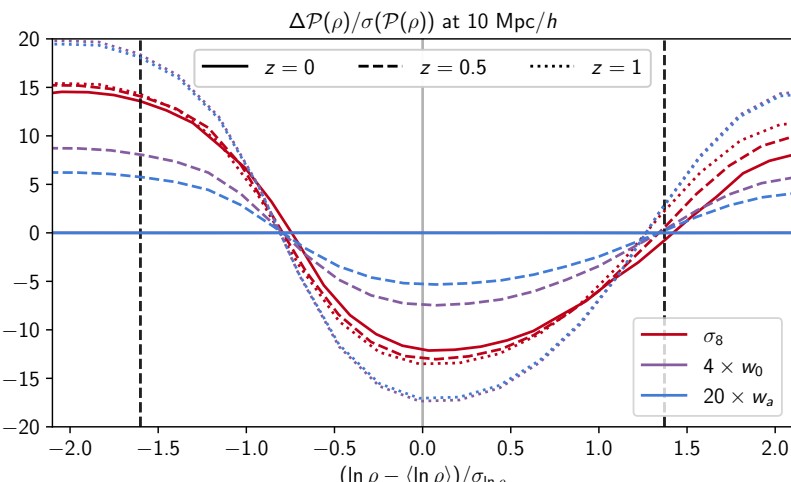

**Figure 4.** Derivatives of the matter PDF in an evolving dark energy universe. The dependence of the matter PDF on $\Omega_m$ is easily distinguished from the others by its distinct skewness (see Figure 3) and hence not shown here. The $\sigma_8$, $w_0$, and $w_a$ derivatives are similar in shape, but have different redshift evolutions, which allows for degeneracy breaking.

### 3.4. Fisher Forecasts for Modified Gravity Detection and Dark Energy Constraints

Figures 5 and 6 show the Fisher forecast constraints for $f(R)$ gravity with $|f_{R0}| = 10^{-6}$ and for $w_0 w_a$CDM about a $\Lambda$CDM fiducial (forecasts for DGP gravity can be found in [15]). In both the modified gravity and the dark energy extended cosmologies, the matter PDF alone provides constraints competitive with the matter power spectrum. More importantly, however, they provide complementary information, demonstrated by the different degeneracy directions. This indicates that the matter PDF is recovering independent non-Gaussian information beyond the power spectrum.

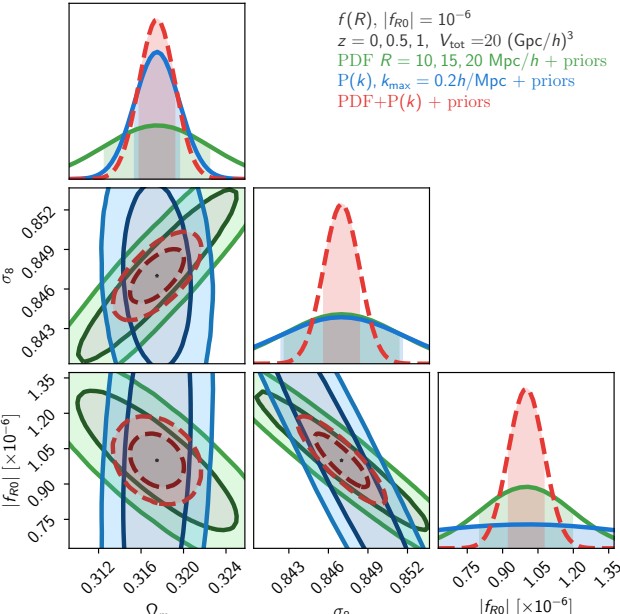

**Figure 5.** Forecast constraints on $f(R)$ gravity using a *Euclid*-like volume. These are marginalised over all other $\Lambda$CDM parameters, and include a prior on $\Omega_b$ and $n_s$ described in [15].

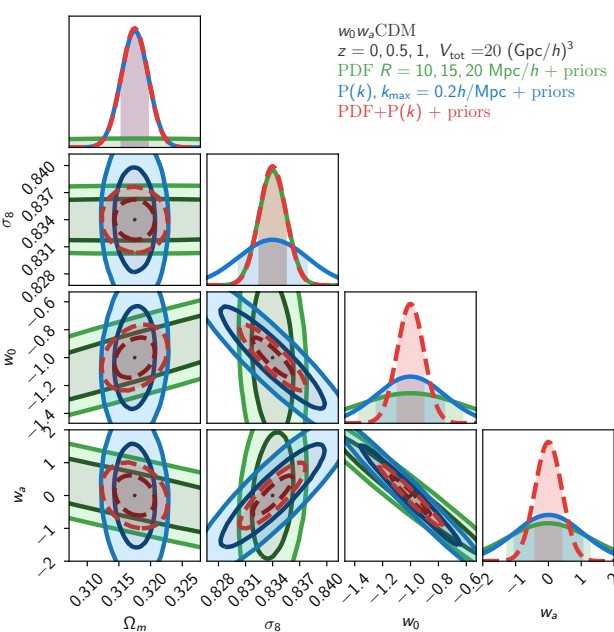

**Figure 6.** Forecast constraints on $w_0 w_a$CDM dark energy using a *Euclid*-like volume. These are marginalised over all other $\Lambda$CDM parameters, and include a prior on $\Omega_b$ and $n_s$ described in [15].

Combining the PDF and the power spectrum allows for a $5\sigma$ detection of both $f(R)$ and DGP gravity (see Table 1), and at least doubles the constraining power for other parameters such as $\sigma_8$ over just power spectrum alone. For evolving dark energy, the improvement is quantified by the dark energy Figure of Merit (FoM), equal to the inverse area of the contour in the $w_0$-$w_a$ plane. Adding PDF information to the power spectrum increases the FoM by a factor of 5 (summarised in Table 2). The resulting FoM is in the range expected to be reached by *Euclid* in combining spectroscopic galaxy clustering and weak lensing [23].

**Table 1.** Detection significance for a fiducial $f(R)$ with $|f_{R0}| = 10^{-6}$ and DGP model with $\Omega_{rc} = 0.0625$. The stronger $f(R)$ constraints are expected from the additional skewness in the PDF response to $|f_{R0}|$ as seen in Figure 3.

|  | $f(R)$ **Detection** | **DGP Detection** |
|---|---|---|
| PDF, 3 scales + prior | $5.15\sigma$ | $1.17\sigma$ |
| $P(k)$, $k_{max} = 0.2$ h/Mpc + prior | $2.01\sigma$ | $2.42\sigma$ |
| PDF + $P(k)$ + prior | $13.40\sigma$ | $5.19\sigma$ |

**Table 2.** Constraints from mildly non-linear scales on $\sigma_8$, $w_0$, and $w_a$ as well as the dark energy Figure of Merit (FoM) coming from the matter PDF, power spectrum, and their combination.

|  | $\frac{\sigma[\sigma_8]}{\sigma_8^{fid}}$ | $\sigma[w_0]$ | $\sigma[w_a]$ | **FoM** |
|---|---|---|---|---|
| PDF, 3 scales + prior | 0.18% | 0.37 | 1.25 | 27 |
| $P(k)$, $k_{max} = 0.2$ h/Mpc + prior | 0.45% | 0.24 | 1.03 | 50 |
| PDF + $P(k)$ + prior | 0.17% | 0.09 | 0.40 | 243 |

## 4. Conclusions

Standard two-point statistics are not sufficient to make full use of the information content in the cosmic large-scale structure, and would leave large amounts of data from current and upcoming galaxy surveys under utilised. The full shape of the matter density PDF in spheres has been shown to provide great complementarity to the standard two-point statistics, and allows extraction of information from the non-linear regime. The analytic

framework described here has been successfully applied to ΛCDM universes along with extensions including primordial non-Gaussianity [24] and massive neutrinos [22]. This work demonstrates that the LDT formalism continues to work in modified gravity and dark energy scenarios , providing a powerful non-Gaussian probe of fundamental physics complementary to two-point statistics.

While the analysis presented here is idealised in that it relies on knowledge of the true matter distribution, it is encouraging for realistic scenarios. In the case of ΛCDM cosmologies, the LDT approach has been translated into several observable quantities, including weak lensing [25–27], galaxy clustering [28,29], and density-split statistics [30,31]. Given the theoretical information content in the matter PDF demonstrated here, extending the LDT framework to observables in the context of modified gravity would be a worthwhile endeavour for constraining both astrophysical (e.g., baryonic feedback, intrinsic alignment, and galaxy bias) and cosmological parameters to complement two-point statistics.

**Author Contributions:** This text was written by A.G. and edited by C.U. All authors have read and agreed to the published version of the manuscript.

**Funding:** A.G. is supported by an EPSRC studentship under Project 2441314 from UK Research & Innovation.

**Data Availability Statement:** The code for computing the matter PDFs, as well as the simulations used can be found in the data availability section of [15].

**Acknowledgments:** The figures in this work were created with Matplotlib [32] and ChainCosumer [33], making use of the NumPy [34] and SciPy [35] Python libraries.

**Conflicts of Interest:** The authors declare no conflict of interest.

## Abbreviations

The following abbreviations are used in this manuscript:

| | |
|---|---|
| ΛCDM | Lambda Cold Dark Matter (model of cosmology) |
| CMB | Cosmic Microwave Background |
| DE | Dark Energy |
| DGP | Dvali-Gabadadze-Porrati (model of gravity) |
| FoM | (Dark Energy) Figure of Merit |
| GR | General Relativity |
| LDT | Large Deviations Theory |
| LSS | Large-Scale Structure (of the universe) |
| MG | Modified Gravity |
| PDF | Probability Distribution Function |

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
