# Peer review of "One-Point Statistics Matter in Extended Cosmologies"

_universe, doi:10.3390/universe8010055_

Round 1
Reviewer 1 Report
The authors of the submission "One-point statistics matter in extended cosmologies" have considered the approach presented in Cataneo et al. 2021 based on using the Large Deviation Theory (LDT) for extended models beyond LCDM. The authors use the LDT for predicting the probability distribution function (PDF) of spheres in the matter field, and extract non-Gaussian information beyond the two-point correlation function for extended models. Using a Fisher forecast for Euclid-like surveys, the authors show that adding the PDF to the standard power spectrum significantly reduces the error bars. Moreover, they show that the PDF contains enough non-Gaussian information to improve the constraints on extended models.
This proceedings submission is based on the results presented in Cataneo et al. 2021 by the same authors and additional collaborators.
The submission is very well written, clear, and with the appropriate references. I only have very few minor comments that I would like to see addressed before giving my recommendation for publication:
- Could the authors clearly mention (at the beginning of the results section, for example), that the results presented in this submission are based on Cataneo et al. 2021, such that the reader know that some might not be totally original from this submission?
- Could the authors clarify why the w0waCDM model is not represented in Fig. 2? Although the w0wa effect is similar to that of DGP gravity, it might be interesting to see both of them in the plot.
- I guess the authors take into account the correlation between the PDF and the power spectrum when building the Fisher matrix with the concatenated data vector. However, could they clarify it?
- The authors mention that the data covariance used in the Fisher matrix was measured from the fiducial LCDM runs of the Quijote simulations. Could the authors discuss what is the impact on the final results if the covariance is for one of the extended models? Do the results change for f(R) if the authors consider an f(R) covariance, instead of a LCDM one?
- The authors mention that the figure-of-merit obtained with the combination of PDF and power spectrum is in the range expected to be reached by Euclid when combining galaxy clustering and weak lensing. I would suggest the authors to clarify that this value corresponds to Euclid's figure-of-merit when combining spectroscopic galaxy clustering with weak lensing. Adding photometric galaxy clustering (with the galaxy-galaxy lensing cross-correlations) will provide a much larger figure-of-merit.
Reviewer 2 Report
The paper is written in a clear and concise form, it presents that the full matter density PDF in spheres are complementary to other types of surveys, and improve the constraints on the cosmological parameters.
Perhaps the only question may be, how the results change when considering a different wave number k, as the authors only used k=0.2.
It is also customary to add the constraints from H0 or ns to complement the analysis.
